# New Biomarkers in the Prognostic Assessment of Acute Heart Failure with Reduced Ejection Fraction: Beyond Natriuretic Peptides

**DOI:** 10.3390/ijms26030986

**Published:** 2025-01-24

**Authors:** Marcelino Cortés, Jairo Lumpuy-Castillo, Camila Sofía García-Talavera, María Belén Arroyo Rivera, Lara de Miguel, Antonio José Bollas, Jose Maria Romero-Otero, Jose Antonio Esteban Chapel, Mikel Taibo-Urquía, Ana María Pello, María Luisa González-Casaus, Ignacio Mahíllo-Fernández, Oscar Lorenzo, José Tuñón

**Affiliations:** 1Cardiology Department, Fundación Jiménez Díaz University Hospital, 28040 Madrid, Spain; lara.miguelg@quironsalud.es (L.d.M.); antonio.bollas@quironsalud.es (A.J.B.); jose.rotero@quironsalud.es (J.M.R.-O.); jose.estebanc@quironsalud.es (J.A.E.C.); mikel.taibo@quironsalud.es (M.T.-U.); ampello@quironsalud.es (A.M.P.); jtunon@quironsalud.es (J.T.); 2Faculty of Medicine and Biomedicine, Universidad Alfonso X el Sabio (UAX), 28691 Madrid, Spain; 3Laboratory of Diabetes and Vascular Pathology, IIS-Fundación Jiménez Díaz, Universidad Autónoma, 28040 Madrid, Spain; jlumpuy@gmail.com (J.L.-C.); olorenzo@fjd.es (O.L.); 4Biomedical Research Network on Diabetes and Associated Metabolic Disorders (CIBERDEM), Carlos III National Health Institute, 28029 Madrid, Spain; 5Cardiology Department, Complejo Hospitalario San Millán—San Pedro, 26004 Logrono, La Rioja, Spain; camila.garcia.talavera@gmail.com; 6Cardiology Department, HM Montepríncipe University Hospital, 28660 Madrid, Spain; belen.arroyo.rivera@gmail.com; 7Department of Laboratory Medicine, La Paz University Hospital, 28046 Madrid, Spain; mlgcasaus@gmail.com; 8Biostatistics and Epidemiology Unit, IIS-Fundación Jiménez Díaz, 28040 Madrid, Spain; imahillo@fjd.es; 9Department of Medicine, Faculty of Medicine, Universidad Autónoma de Madrid, 28049 Madrid, Spain; 10Biomedical Research Network on Cardiovascular Diseases CIBERCV, Carlos III National Health Institute, 28029 Madrid, Spain

**Keywords:** systolic heart failure, biomarkers, prognosis, GDF 15, sST2

## Abstract

Natriuretic peptides are established biomarkers related to the prognosis of heart failure. New biomarkers have emerged in the field of cardiovascular disease. The prognostic value of these biomarkers in heart failure with reduced left ventricular ejection fraction is not well-established. We conducted a prospective, single-centre study, including (July 2019 to March 2023) 104 patients being consecutively admitted with a diagnosis of acute heart failure with reduced ejection fraction decompensation. The median follow-up was 23.5 months, during which 20 deaths (19.4%) and 21 readmissions for heart failure (20.2%) were recorded. Plasma biomarkers, such as NT-proBNP, GDF-15, sST2, suPAR, and FGF-23, were associated with an increased risk of all-cause mortality. However, a Cox regression analysis showed that the strongest predictors of mortality were an estimated glomerular filtration rate (HR 0.96 [0.93–0.98]), GDF-15 (HR 1.3 [1.16–1.45]), and sST2 (HR 1.2 [1.11–1.35]). The strongest predictive model was formed by the combination of the glomerular filtration rate and sST2 (C-index 0.758). In conclusion, in patients with acute decompensated heart failure with reduced ejection fraction, GDF-15 and sST2 showed the highest predictive power for all-cause mortality, which was superior to other established biomarkers such as natriuretic peptides. GDF-15 and sST2 may provide additional prognostic information to improve the prognostic assessment.

## 1. Introduction

Heart failure (HF) remains a prevalent and relevant health problem today. It is estimated that approximately 1–2% of the adult population suffers from HF, reaching a prevalence of over 10% in elderly patients [1,2]. Despite major advances in the treatment and management of these patients in recent years, the mortality and morbidity associated with HF remains high [3]. Several markers and prognostic models have been studied over the last decades in order to predict which patients are at increased risk of events [4]. Among these risk markers, biomarkers, elements detectable in analytical samples, stand out. Their prognostic and diagnostic role has been analysed in the cardiovascular field, and also specifically in the field of HF [5]. Markers have been described at the neurohormonal level, including inflammatory mediators and cell damage, among others, with natriuretic peptides standing out in particular. They have been fully implemented in clinical practice, playing a prognostic role, guiding treatment, or even being involved in the very definition of HF [6]. In recent years, new biomarkers(such as those related to inflammation, oxidative stress, tissue damage, and renal function) have been sought to provide new advances in the management of patients with HF. To date, these new biomarkers have not been successfully used in routine clinical practice [7]. However, some of them, such as soluble Suppression of Tumorigenicity 2 (sST2), Growth Differentiation Factor-15 (GDF-15), soluble urokinase Plasminogen Activator Receptor (suPAR), Fatty Acid Binding Protein 4 (FABP4), or mineral metabolism (MM) biomarkers (Fibroblast Growth Factor 23 (FGF23), klotho, phosphorus (P), parathyroid hormone (PTH), or 1-25-dihydroxyvitamin D (calcidiol) have shown promising results in relation to the diagnosis and prognosis of HF.

The aim of our study was to analyse the prognostic role of these new biomarkers in HF with reduced ejection fraction (HFrEF) in the setting of discharge after admission for acute heart failure, assessing and comparing the prognostic power of these biomarkers and their associations, as well as their added value to natriuretic peptides.

## 2. Results

### 2.1. Baseline Characteristics of Patients

We included 104 patients in our study (Figure 1). The median age of our population was 66.7 years, mostly comprised of male patients (78.8%). The percentage of patients with comorbidities was relatively high. Thus, 29.8% had chronic lung disease (COPD, asthma, OSA), 31.7% had chronic kidney disease, 10.6% had a history of stroke, and 30.8% of patients were in atrial fibrillation at the time of inclusion. The percentage of diabetics in our population reached almost 50%, and more than 66% of patients were hypertensive. In 31% of the study population, the main underlying cause of LV systolic dysfunction was ischaemic heart disease, with 27.9% of patients having a history of previous STEMI. After hospital discharge, patients were followed up in the heart failure unit (HFU) according to the study protocol, achieving treatment rates with BB greater than 90%, ARBS-ACEIS-ARNI 87%, MRAs 74%, and SGLT2i 72.1%. Figure 1 represents treatment in the study population (at the end of follow-up).

Following the described methodology, we analysed plasma samples obtained from our study population at admission. Table 1 presents the results of the principal biochemical blood parameters (renal function, iron profile, haemogram, and others) in our study population. It also shows the results of the wide range of biomarkers determined in our study: the most classical ones (CK-MB, NT-proBNP, TnI), as well as a wide representation of new biomarkers that have shown a potential prognostic role in cardiovascular disease according to several studies performed in recent years (mineral metabolism biomarkers [calcidiol, P, FGF23, klotho, PTH], inflammatory and immune processes biomarkers [GDF-15, sST2, suPAR, C-reactive protein], lipid metabolism [FABP4], and atrial peptides [NT-proANP]).

All patients were followed up in the HFU of our centre. After a median follow-up of 23.5 months, 20 deaths and 21 heart failure readmissions were recorded. Figure 2 represents the Kaplan–Meier curves in our study population with regards to all-cause death and readmissions for heart failure.

### 2.2. Association of Biomarkers and All-Cause Death

At the end of the follow-up period of our study population, 20 deaths were recorded in our population. Regarding the cause of death, seven deaths were related to a cardiovascular event (including three sudden deaths) and eight were due to a non-cardiac cause. In the remaining five patients, the origin of death could not be determined. Table 1 and Figure 1 show comparatively different variables (clinical, treatment, and biochemical parameters) with respect to all-cause mortality. Variables such as age, CKD, previous cancer, previous admissions for HF, or advanced functional class were associated with higher mortality in the univariate study. Treatment with SGLT2i was shown to be a protective factor, with a significantly lower rate of use in patients who died. In terms of biochemical parameters, glomerular filtration rate and haemoglobin were associated with total mortality, as expected. As for biomarkers, several of them were associated with worse prognosis in our study population. Higher levels of C-reactive protein, NT-proBNP, GDF-15, sST2, and suPAR were associated with an increased risk of mortality. Regarding biomarkers of mineral metabolism, FGF-23 was also associated with an increased risk of all-cause mortality, with a borderline significant relationship with calcium.

As described in the methodology, we designed multivariable predictive models for all-cause mortality considering, for the selection of variables, those that showed a C index 0.7 in the univariable Cox regression analysis (Figure 3). Following this methodology, we found three variables with adequate predictive power: glomerular filtration rate, GDF-15, and sST2. These three variables showed greater predictive power than the rest of the clinical and biochemical variables. We used these three variables to generate different predictive models of mortality by combining them. In this way, three predictive models could be generated. The model combining GDF-12 and sST2 showed adequate predictive power (C-index 0.744), although the most powerful model resulted from the combination of sST2 and the estimated glomerular filtration rate [C-index 0.758. The equation for the model is: Ŝ(t; eGFR, sST2) = Ŝ_0_(t)^exp(−0.034eGFR+0.013sST2)^]. Figure 4 shows, comparatively, the different predictive models for mortality obtained in our analysis.

### 2.3. Hospital Readmissions for Heart Failure

At the end of the follow-up period of our study population, there were 21 patients with HF readmissions. After univariate survival analysis using the Cox regression, biomarkers such as GDF-15, suPAR, calcidiol, and FGF23 were associated with readmissions. Other variables, such as advanced NYHA functional class (NYHA III or IV), HF admissions prior to study inclusion, or previous history of coronary revascularisation were significantly associated with HF readmissions as well. However, none of these variables achieved sufficient predictive ability according to the statistical methodology described, with a C-index in all cases of less than 0.7. Table 2, Table 3 and Table 4 show the results of the statistical analyses of our study population with respect to readmissions for heart failure.

## 3. Discussion

HF is a clinical syndrome of marked relevance today, with a high prevalence and incidence [6,8,9]. Mortality and morbidity associated with HF remain significant, with a mortality rate of around 8% per year and a one-year hospitalization rate of over 28% according to some registries [10]. Adequately identifying patients with worse prognoses using different prognostic markers allows for the selection of patients with the greatest care needs, thus allowing for a more rational management of health system resources.

### 3.1. Prognostic Markers in Heart Failure: Natriuretic Peptides

Several prognostic markers and models have been evaluated in recent decades within HFrEF [4]. Among these risk markers are biomarkers. The prognostic and diagnostic role of these biomarkers has been analysed in different cardiovascular diseases, including specifically in HF [5,11]. The most widely used in routine clinical practice are natriuretic peptides, having shown utility in the diagnosis, risk stratification, and clinical follow-up of patients with HF [12,13]. However, natriuretic peptides have some limitations. Their blood levels are influenced by several factors, like age, renal failure, hypertrophy, or obesity [14,15]. Moreover, natriuretic peptides are produced almost exclusively in the heart, in response to increased end-diastolic wall stress in the left ventricle [16], so their blood levels are determined solely by this condition.

### 3.2. New Biomarkers in Heart Failure: Inflammation

There is growing evidence that HF is a much more complex clinical syndrome, with diverse aetiologies and pathophysiological mechanisms involved, including inflammatory and immunomodulatory processes not measurable by natriuretic peptides [17,18]. For these and other reasons, in recent years, several studies have evaluated the role of new biomarkers that may add diagnostic and prognostic value to natriuretic peptides [7]. In our work, we have collected some of these promising new biomarkers and analysed their prognostic role in the setting of discharge after admission for HFrEF. Our analysis shows a significant relationship of NT-proBNP with mortality, but also other biomarkers, such as CRP, GDF-15, sST2, suPAR, or FGF-23 (related to these inflammatory and immunomodulatory processes). Moreover, according to our results, the predictive power of sST2 and GDF-15 was superior to other biomarkers (including natriuretic peptides), leading to more powerful predictive models (in association with the estimated glomerular filtration rate).

### 3.3. GDF-15 and sST2

GDF-15 and sST2 are biomarkers belonging to the TGF-β and interleukin-1 receptor families, respectively [19,20]. In situations of myocardial stress or cellular overload, they are highly expressed in cardiomyocytes, but also in other cell types. In addition, they are also associated with different pathophysiological conditions, such as oxidative stress, hypoxia, tissue injury, and inflammatory and immune processes [21,22,23]. Several publications have shown a prognostic relationship of these biomarkers with cardiovascular disease [24,25,26,27,28,29,30], and specifically with HF. In this setting, increased levels of GDF-15 have been found in patients with HF [31], as well as an increased risk of developing HF [32]. Several studies have shown a worse prognosis in patients with chronic stable HFrEF and elevated levels of GDF-15 or sST2 [33,34,35,36,37,38,39,40,41], even with a stronger prognostic power than other more traditional variables, including natriuretic peptides [42]. However, in the setting of acute HF in patients with HFrEF, evidence is scarce. Although several studies have been published showing the prognostic value of these biomarkers in acute HF, most of them are based on a very heterogeneous population, analysing HFpEF and HFrEF together, not differentiating both entities [23,43,44,45,46,47,48,49], or with HFrEF criteria different from current recommendations [50]. In contrast to these publications, we focused on a specific and homogeneous population of patients with decompensated HFrEF, providing a greater robustness to our results in relation to this subgroup of patients. This subgroup has a particularly poor prognosis, as demonstrated by the high mortality in our study group. Our results show an important prognostic role of GDF-15 and sST2: allowing the identification of those patients with a higher risk and facilitating a better allocation of resources. In these patients with a worse outcome, they could benefit from therapeutic intensification and/or closer clinical follow-up, facilitating clinical decision-making regarding specific therapies or programmes. This could result in a clinical benefit, improving patient outcomes. However, specific studies with biomarker-guided therapy and follow-up would be needed to confirm this.

### 3.4. Other Biomarkers Analysed in Our Study

We analysed other biomarkers that, in recent years, have been related to cardiovascular disease, such as suPAR, FABP4, and MM biomarkers (P, PTH, vitamin D, FGF-23, klotho). In this setting, changes in the different components of the MM cascade have been associated with cardiac alterations (functional and structural) and heart diseases, playing a prognostic role for even the general population and uncertain CVD [51,52,53,54,55,56]. Specifically, alterations of several MM biomarkers have been associated with an increased incidence of HF [57,58,59,60,61,62,63,64,65], as with suPAR [66] and FABP4 [67]. Some of these biomarkers have demonstrated a prognostic role in HF, including in HFrEF [68,69,70,71,72,73,74]. However, there are little or no data on the prognostic role of these biomarkers in acute HF, and, generally, they do not differentiate between HFrEF and HFpEF [75,76]. In our study population of patients with acute HFrEF, only FGF-23 and suPAR showed a statistically significant relationship with prognosis, losing their significance in multivariate analysis. It is possible that a larger study population could change our results regarding these biomarkers.

In summary, results such as those obtained in our population of patients with decompensated HFrEF, together with those published by other authors in other populations of patients with HF, support the prognostic utility of these new biomarkers (specifically sST2 and GDF-15). HF is a complex clinical syndrome, with various pathophysiological mechanisms involved that are reflected in these new biomarkers (immune processes, inflammation, tissue injury etc.). Their use could provide additional prognostic information that could improve the prognostic assessment of our patients with HF.

## 4. Materials and Methods

### 4.1. Patients and Study Design

We carried out a single-centre, observational prospective study. Between July 2019 and March 2023, patients admitted to our centre with a principal diagnosis of decompensated HFrEF were consecutively included. Inclusion criteria were as follows: (1) diagnosis prior to or during admission of HFrEF, according to the 2021 recommendations of the European Society of Cardiology (symptoms and signs of HF and LVEF < 40%) [6]; (2) HF as the main cause for admission; and (3) referral at discharge to the HFU of our centre for follow-up. Exclusion criteria for the study, as well as for follow-up of patients in the HFU, were: (a) HFrEF due to heart disease potentially reversible with cardiac surgery or programmed short-term intervention (such as revascularization, surgical valve replacement-repair, percutaneous aortic prosthesis implantation, or mitral valvuloplasty); (b) non-cardiac end-stage disease with life expectancy of less than 6 months; (c) decompensation of HF secondary to non-cardiac cause; (d) very advanced heart failure (INTERMACS classes1 to 5); and (e) patients expected to be unable to follow the protocol.

During admission, several clinical and demographic variables were collected from the included patients. After patients gave their consent to be included in the study, blood samples were drawn after 12 h of fasting. Blood sampling was performed as soon as possible after the patient’s admission date. These venous blood samples were collected in tubes with and without EDTA and were centrifuged at 2500× *g* for 10 min. The obtained plasma samples were stored in 2 mL cryovials at −80 °C. After hospital discharge, all patients were referred to the HFU of our hospital and included in the specific follow-up programme of this unit. This programme included follow-up visits by both physicians and specialised nurses, with early visits after discharge, as well as repeated medical check-ups throughout the follow-up, according to the patient’s needs. During this follow-up, patients were clinically assessed, medical treatment was optimised, and specific patient education activities, among other actions, were carried out. During patient follow-up in the HFU, several clinical and follow-up variables were collected for further analysis. Figure 5 summarises the methodology of our study.

This investigation was carried out in accordance with the principles outlined in the Declaration of Helsinki. Written informed consent was obtained from all participants. Moreover, the study design and protocol were approved by the Clinical Research Ethics Committee of our institution (Ref. PIC157-18_FJD).

### 4.2. Clinical Outcomes

The outcomes analysed in our study were the rate of all-cause death and admission due to HF. HF admission was defined as admission to a healthcare facility lasting >24 h due to the worsening of HF symptoms and followed by specific treatment for HF (regardless of the cause of decompensation). Clinical events and death during follow-up were collected from patients’ electronic health records or, if not available, from telephone interviews with patients or relatives.

### 4.3. Biochemical Analysis

Serum and plasma samples were collected and stored (at −80 °C) during hospital admission (with consent of patients in the study). We measured the usual blood parameters (complete blood count, lipid profile, kidney function, liver function, etc.). Additionally, we analysed the levels of several specific biomarkers. The plasma concentrations of human GDF-15, sST2, and suPAR were measured using the automated immunoassay system ELLA from Protein Simple (Bio-Techne, MN, USA), following the manufacturer’s instructions. The detection kits used were SPCKB-PS-000269 (GDF-15), SPCKB-PS-000221 (sST2), and SPCKB-PS-007370 (suPAR). Each plasma sample was run in triplicate, and the inter-plate coefficient of variation (CV%) was less than 4% in all cases. Also, plasma levels of human NT-ProANP and FABP4 were measured by immunoassay using Quantikine^®^ colorimetric sandwich ELISA kits (ref: DANP00 and DFBP40, respectively) from R&D Systems (R&D Systems, Inc., Minneapolis, MN, USA). The absorbance was set at 450 nm with a wavelength correction at 570 nm using a plate reader (EnSpire^®^ Multimode Reader, Perkin Elmer, Waltham, MA, USA). For both assays, the intra-assay CV% was less than 4.5%, and the inter-plate CV was less than 7.5%. Additionally, the creatine kinase–myocardial band (CK-MB) levels were measured by immunoassay using VITROS Immunodiagnostic products (CK-MB reagent pack, ref: 1896836, VITROS Immunodiagnostic, Raritan, NJ, USA) at the Analytical Service of the Fundación Jiménez Díaz. For MM biomarkers, plasma calcidiol levels were quantified by chemiluminescent immunoassay (CLIA) on the LIAISON XL analyser (LIAISON 25OH-Vitamin D Total Assay, Dia Sorin, Saluggia, Italy). FGF-23 was measured by enzyme-linked immunosorbent assay (ELISA) recognizing epitopes within the carboxyl-terminal portion of FGF23 (Human FGF23, C-Term, Immutopics Inc., San Clemente, CA, USA). Klotho levels were measured by ELISA (Human Soluble Alpha Klotho Assay Kit, Immuno-Biological Laboratories Co., Hokkaido, Japan). Finally, intact PTH was analysed using a second-generation automated chemiluminescent method (Elecsys 2010 platform, Roche Diagnostics, Mannheim, Germany).

### 4.4. Statistical Analysis

Qualitative variables were presented as absolute and relative frequencies. Associations between qualitative variables were assessed using the Chi-squared test or Fisher’s exact test. Subsequently, the relative risk (RR) was calculated. On the other hand, quantitative variables were described using medians and interquartile ranges (IQR), and comparisons were performed with the Mann–Whitney U test for independent samples. Subsequently, relationships between variables were explored using both univariable and multivariable Cox regression models. Initially, univariable Cox regression analysis was conducted to identify variables associated with all-cause mortality and HF admissions. For each variable, the hazard ratio with its 95% confidence interval, *p*-value, and C-statistic (C-Index) were reported, with the latter being derived through the Leave-One-Out Cross-Validation method. This method was employed to select variables generating univariable models with the best predictive capacity (C-index ≥ 0.7) [77]. A multivariable Cox regression analysis was then performed to identify significant predictors. All statistical analyses were conducted using the Statistical Package for the Social Sciences (SPSS v.26.0, IBM, Armonk, NY, USA), R statistical language version 4.0.5 (R Foundation for Statistical Computing, Vienna, Austria), and the statistical package for the biomedical sciences (MedCalc v.23.0.2, Ostend, Belgium; https://www.medcalc.org, accessed on 1 September 2024).

## 5. Conclusions

In our population of patients with acute heart failure and HFrEF, GDF-15 and sST2 showed the highest predictive power for all-cause mortality, superior to more established biomarkers (natriuretic peptides). Their use would provide additional prognostic information and could improve the prognostic assessment of our acute HF patients.

## Figures and Tables

**Figure 1 ijms-26-00986-f001:**
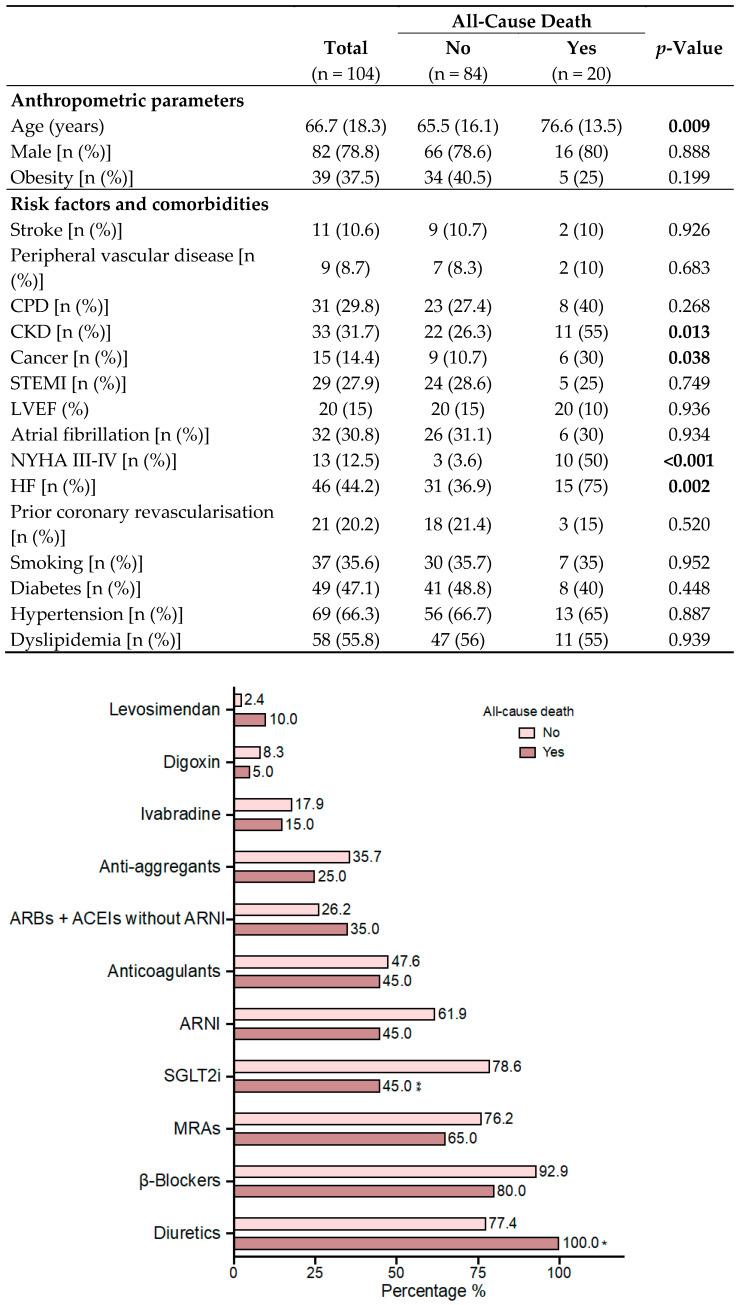
Comparison of baseline characteristics (clinical and treatment) according to clinical endpoint during follow-up (all-cause death). ACEI: angiotensin converting enzyme inhibitor; ARB: angiotensin receptor blocker; ARNI: angiotensin receptor/neprilysin inhibitor; CPD: chronic pulmonary disease; CKD: chronic kidney disease; HF: admission for heart failure prior to inclusion; LVEF: left ventricular ejection fraction; MRAs: mineralocorticoid receptor antagonists; SGLT2i: sodium-glucose co-transporter-2 inhibitors; STEMI: ST-elevation myocardial infarction. Bold *p*-values and asterisks indicate statistical significance.

**Figure 2 ijms-26-00986-f002:**
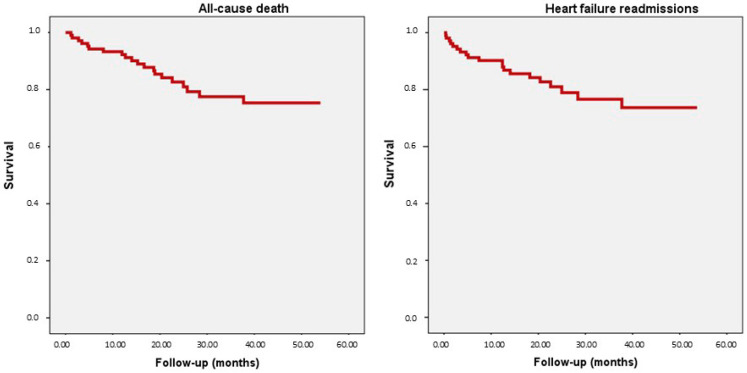
All-cause death and heart failure readmissions: Kaplan–Meier curves.

**Figure 3 ijms-26-00986-f003:**
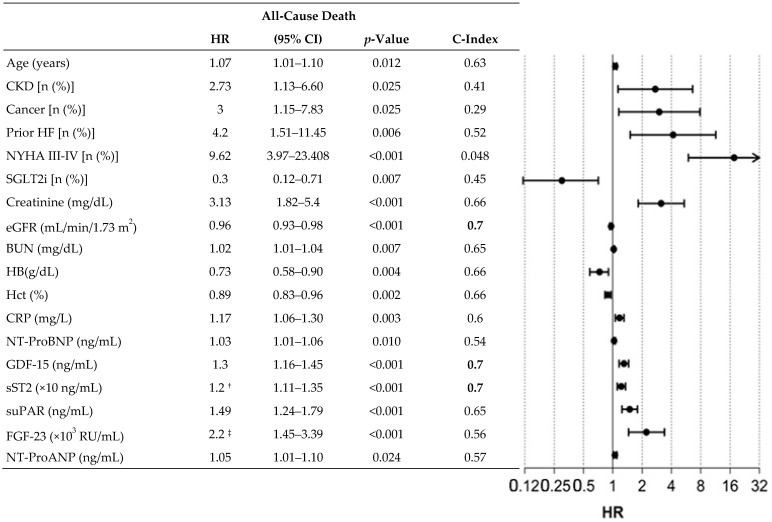
All-cause mortality: univariate Cox regression analysis (statistically significant variables). CKD: chronic kidney disease; CRP: C-reactive protein; eGFR: estimated glomerular filtration rate; FGF-23: Fibroblast Growth Factor 23; GDF-15: Growth Differentiation Factor-15;HB: haemoglobin; HR: Hazard Ratio; prior HF: admission for heart failure prior to inclusion; Hct: haematocrit; NT-ProANP: N-terminal Proatrial Natriuretic Peptide; NT-ProBNP: N-terminal Probrain Natriuretic Peptide; sST2: soluble Suppression of Tumorigenicity 2; suPAR: soluble urokinase Plasminogen Activator Receptor. ^†^ HR indicates change per 10 units. ^‡^ HR indicates change per 1000 units. Bold C-index values indicate the best predictive capacity.

**Figure 4 ijms-26-00986-f004:**
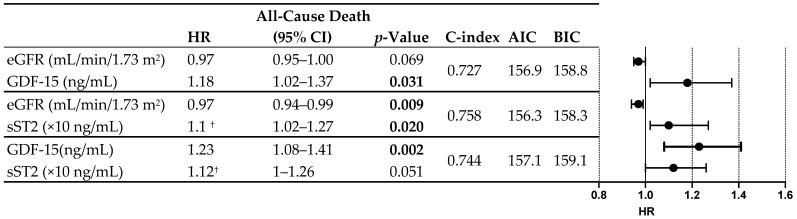
All-cause mortality: multivariate Cox regression analysis and predictive models. AIC: Akaike Information Criterion; BIC: Bayesian Information Criterion; eGFR: estimated glomerular filtration rate; GDF-15: Growth Differentiation Factor-15; sST2: soluble Suppression of Tumorigenicity 2. Bold *p*-values indicate statistical significance. ^†^ HR indicates change per 10 units.

**Figure 5 ijms-26-00986-f005:**
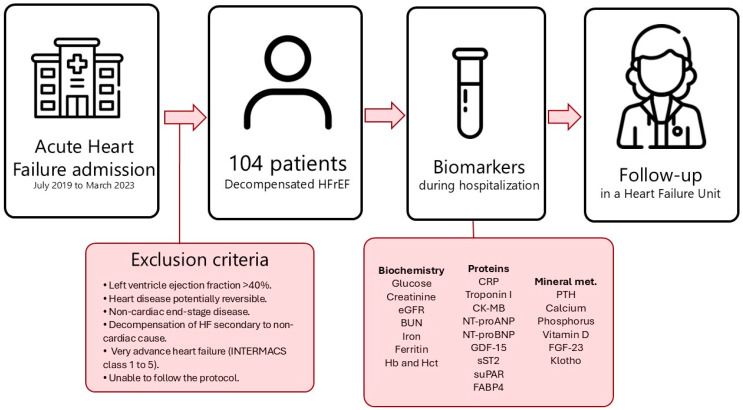
Flow chart of the stages of the study.

**Table 1 ijms-26-00986-t001:** Comparison of baseline characteristics (biochemical analysis) according to clinical endpoint during follow-up (all-cause death).

		All-Cause Death		
	Total	No	Yes	*p*-Value
	(n = 104)	(n = 84)	(n = 20)	
**Biochemistry**				
Glucose (mg/dL)	113 (45)	111.5 (45)	117.5 (49)	0.954
Creatinine (mg/dL)	1.1 (0.6)	1 (0.4)	1.5 (1)	**<0.001**
eGFR (mL/min/1.73 m^2^)	66.9 (38)	70.3 (36.7)	46 (27.9)	**<0.001**
BUN (mg/dL)	25 (16)	23.5 (14)	38.5 (26)	**0.03**
Serum iron level (µg/dL)	54 (37.8)	54 (39)	48 (42.5)	0.615
Ferritin (ng/mL)	147.4 (220)	137.5 (231)	163 (183)	0.961
HB (g/dL)	13.6 (3.6)	13.9 (3.2)	11.8 (3)	**0.006**
Hct (%)	41.9 (9.4)	43 (7.9)	36.9 (9.9)	**0.008**
**ProteinBiomarkers**				
CRP (mg/L)	0.96 (2.4)	0.9 (2)	2.6 (4.6)	**0.027**
TnI (ng/mL)	0.04 (0.1)	0.04 (0.07)	0.04 (0.08)	0.834
CK-MB (ng/mL)	1.1 (0.7)	0.99 (1.4)	1.12 (1.4)	0.091
NT-proBNP (ng/mL)	6.4 (10.7)	6.1 (8.7)	10.1 (14.5)	**0.029**
NT-proANP (ng/mL)	29.7 (10)	28.9 (11.4)	31.8 (6.8)	0.175
GDF-15 (ng/mL)	3.1 (2.4)	2.9 (2.1)	5 (6.4)	**<0.001**
sST2 (×10 ng/mL)	3.53 (3.5)	3.09 (2.9)	5 (5.82)	**<0.001**
suPAR (ng/mL)	2.9 (1.5)	2.8 (1.4)	3.5 (2.1)	**0.004**
FABP4 (ng/mL)	44.21 (32.6)	43.2 (32.2)	50 (54.2)	0.152
**MM Biomarkers**				
PTH (pg/mL)	71 (49.5)	67.5 (46)	85 (80)	0.416
Calcium (mg/dL)	9.4 (0.8)	9.4 (0.9)	9.6 (0.6)	**0.048**
Phosphorus (mg/dL)	3.7 (1)	3.7 (1)	3.6 (1.3)	0.948
25(OH)D (ng/mL)	24.5 (27.2)	25.5 (26.5)	19.3 (22.2)	0.345
FGF-23 (×10^3^ RU/mL)	0.36 (0.5)	0.33 (0.4)	0.90 (1.8)	**0.034**
Klotho (pg/mL)	458.5 (242)	458.5 (235)	461 (264)	0.603

25(OH)D: 1-25-dihydroxyvitamin D;eGFR: estimated glomerular filtration rate; FABP4: Fatty Acid Binding Protein 4; FGF-23: Fibroblast Growth Factor 23; GDF-15: Growth Differentiation Factor-15; HB: haemoglobin; Hct: haematocrit; CK-MB: creatine kinase-MB; CRP: C-reactive protein; NT-ProANP: N-terminal Proatrial Natriuretic Peptide; NT-ProBNP: N-terminal Probrain Natriuretic Peptide; PTH: parathormone; sST2: soluble Suppression of Tumorigenicity 2; TnI: troponin I; suPAR: soluble urokinase Plasminogen Activator Receptor. Bold *p*-values indicate statistical significance.

**Table 2 ijms-26-00986-t002:** Baseline characteristics: clinical and treatment. Comparison according to heart failure readmission.

		Heart Failure Readmission		
	Total	No	Yes	*p*-Value
	(n = 104)	(n = 83)	(n = 21)	
**Anthropometric parameters**				
Age (years)	66.7 (18.3)	66.7(20.1)	64.8 (12.14)	0.310
Male [n (%)]	82 (78.8)	66 (79.5)	16 (76.2)	0.739
Obesity [n (%)]	39 (37.5)	30 (36.1)	9 (42.9)	0.570
**Risk factors and comorbidities**				
Stroke [n (%)]	11 (10.6)	8 (9.6)	3 (14.3)	0.691
Peripheral vascular disease [n (%)]	9 (8.7)	6 (7.2)	3 (14.3)	0.381
CPD [n (%)]	31 (29.8)	22 (26.5)	9 (42.9)	0.183
CKD [n (%)]	33 (31.7)	23 (27.7)	10 (47.6)	0.080
Cancer [n (%)]	15 (14.4)	14 (16.9)	1 (4.8)	0.295
STEMI [n (%)]	29 (27.9)	20 (24.1)	9 (42.9)	0.087
LVEF (%)	20 (15)	20 (15)	20 (10)	0.953
Atrial fibrillation [n (%)]	32 (30.8)	23 (27.7)	9 (42.9)	0.179
NYHA III-IV [n (%)]	13 (12.5)	4 (4.8)	9 (42.9)	**<0.001**
HF [n (%)]	46 (44.2)	29 (34.9)	17 (81)	**<0.001**
Prior coronary revasc. [n (%)]	21 (20.2)	12 (14.5)	9 (42.9)	**0.012**
Smoking [n (%)]	37 (35.6)	28 (33.7)	9 (42.9)	0.435
Diabetes [n (%)]	49 (47.1)	39 (47)	10 (47.6)	0.959
Hypertension [n (%)]	69 (66.3)	55 (66.3)	14 (66.7)	0.972
Dyslipidemia [n (%)]	58 (55.8)	49 (59)	9 (42.9)	0.182
**Pharmacology**				
Anticoagulants [n (%)]	49 (47.1)	36 (43.4)	13 (61.9)	0.129
Anti-agregants [n (%)]	35 (33.7)	28 (33.7)	7 (33.3)	0.972
MRAs [n (%)]	77 (74)	61 (73.5)	16 (76.2)	0.801
SGLT2i [n (%)]	75 (72.1)	62 (74.7)	13 (61.9)	0.243
ARBs + ACEIs without ARNI	29 (27.9)	25 (30.1)	4 (19)	0.312
β-Blockers [n (%)]	94 (90.4)	75 (90.4)	19 (90.5)	0.987
Diuretics [n (%)]	85 (81.7)	66 (79.5)	19 (90.5)	0.350
Digoxin [n (%)]	8 (7.7)	7 (8.4)	1 (4.8)	0.573
Ivabradine [n (%)]	18 (17.3)	16 (19.3)	2 (9.5)	0.518
Levosimendan [n (%)]	4 (3.8)	2 (2.4)	2 (9.5)	0.181
ARNI [n (%)]	61 (58.7)	51 (61.4)	10 (47.6)	0.250

ACEI: Angiotensin converting enzyme inhibitor; ARB: angiotensin receptor blocker; ARNI: angiotensin receptor/neprilysin inhibitor; CPD: chronic pulmonary disease; CKD: chronic kidney disease; HF: admission for heart failure prior to inclusion; LVEF: left ventricular ejection fraction; MRAs: mineralocorticoid receptor antagonists; SGLT2i: sodium-glucose co-transporter-2 inhibitors; STEMI: ST-elevation myocardial infarction. Bold *p*-values indicate statistical significance.

**Table 3 ijms-26-00986-t003:** Baseline characteristics: biochemical analysis. Comparison according to heart failure readmission.

		HF Readmission		
	Total	No	Yes	*p*-Value
	(n = 104)	(n = 83)	(n = 21)	
**Biochemistry**				
Glucose (mg/dL)	113 (45)	113 (35)	99 (73)	0.489
Creatinine (mg/dL)	1.1 (0.6)	1.1 (0.49)	1.2 (0.64)	**0.047**
eGFR (mL/min/1.73 m^2^)	66.9 (38)	68 (35.9)	54 (37.83)	0.111
BUN (mg/dL)	25 (16)	25 (15)	29 (19)	0.395
Serum iron level (µg/dL)	54 (37.8)	54 (41.5)	47 (28)	0.672
Ferritin (ng/mL)	147.4 (220)	137.6 (265)	127 (143)	0.101
HB (g/dL)	13.6 (3.6)	13.7 (3.3)	13 (4.05)	0.709
Hct (%)	41.9 (9.4)	42.5 (8.9)	40 (12.8)	0.755
**ProteinBiomarkers**				
CRP (mg/L)	0.96 (2.4)	0.92 (2.64)	0.99 (2.08)	0.288
TnI (ng/mL)	0.04 (0.1)	0.04 (0.07)	0.05 (0.1)	0.893
CK-MB (ng/mL)	1.1 (0.7)	1.01 (0.75)	1.05 (0.87)	0.929
NT-proBNP (ng/mL)	6.4 (10.7)	7.61 (10.96)	5.08 (5.35)	0.195
NT-proANP (ng/mL)	29.7 (10)	29.69 (9.84)	28.57 (13.71)	0.442
GDF-15 (ng/mL)	3.1 (2.4)	3 (2.25)	4.04 (3.23)	0.072
sST2 (×10 ng/mL)	3.53 (3.5)	3.37 (3.05)	3.98 (3.86)	0.229
uPAR (ng/mL)	2.9 (1.5)	2.8 (1.41)	3.18 (1.4)	0.093
FABP4 (ng/mL)	44.21 (32.6)	44.36 (33.99)	52.95 (29.17)	0.574
**MM Biomarkers**				
PTH (pg/mL)	71 (49.5)	71 (54)	71 (55)	0.156
Calcium (mg/dL)	9.4 (0.8)	9.4 (0.95)	9.5 (0.95)	0.810
Phosphorus (mg/dL)	3.7 (1)	3.6 (1)	3.9 (1.05)	0.305
25(OH)D (ng/mL)	24.5 (27.2)	23 (21.3)	34 (36)	0.211
FGF-23 (×10^3^ RU/mL)	0.36 (0.5)	0.32 (0.36)	0.71(1.58)	0.104
Klotho (pg/mL)	458.5 (242)	452 (230)	529 (278)	0.135

25(OH)D: 1-25-dihydroxyvitamin D; CK-MB: creatine kinase-MB; CRP: C-reactive protein; eGFR: estimated glomerular filtration rate; FABP4: Fatty Acid Binding Protein 4; FGF-23: Fibroblast Growth Factor 23; GDF-15: Growth Differentiation Factor-15; HB: haemoglobin; Hct: haematocrit; NT-ProANP: N-terminal Proatrial Natriuretic Peptide; NT-ProBNP: N-terminal Probrain Natriuretic Peptide; PTH: parathormone; sST2: soluble Suppression of Tumorigenicity 2; TnI: troponin I; suPAR: soluble urokinase Plasminogen Activator Receptor. Bold *p*-values indicate statistical significance.

**Table 4 ijms-26-00986-t004:** Heart failure readmission: univariate Cox regression analysis (statistically significant variables).

	HF Readmission			
	HR	(95% CI)	*p*-Value	C-Index
Creatinine (mg/dL)	2.20	1.14–4.22	0.018	0.58
GDF-15 (ng/mL)	1.22	1.07–1.38	0.003	0.59
suPAR (ng/mL)	1.41	1.12–1.77	0.003	0.60
Calcidiol (ng/mL)	1.02	1.01–1.04	0.006	0.53
FGF-23 (×10^3^ RU/mL)	2.12 ^†^	1.36–3.33	0.001	0.53
CKD [n (%)]	2.40	1.02–5.67	0.046	0.37
HF [n (%)]	7.38	2.47–22.0	<0.001	0.56
NYHA III-IV [n (%)]	12.0	4.58–31.3	<0.001	0.51
Prior coronaryrevasc. [n (%)]	3.43	1.44–8.15	0.005	0.40

CKD: chronic kidney disease; FGF-23: Fibroblast Growth Factor 23; GDF-15: Growth Differentiation Factor-15; HF: admission for heart failure prior to inclusion; suPAR: soluble urokinase Plasminogen Activator Receptor. ^†^ HR indicates change per 1000 units.

## Data Availability

Dataset available on request from the authors. The raw data supporting the conclusions of this article will be made available by the corresponding author (Marcelino Cortés (mcortesg@quironsalud.es)) on request.

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
