# Peer review of "New Biomarkers in the Prognostic Assessment of Acute Heart Failure with Reduced Ejection Fraction: Beyond Natriuretic Peptides"

_ijms, 2025, doi:10.3390/ijms26030986_

Round 1
Reviewer 1 Report
Comments and Suggestions for Authors
Cortés et al submitted an interesting manuscript entitled “New biomarkers in the prognostic assessment of acute heart failure with reduced ejection fraction: beyond natriuretic peptides” in which the authors evaluated certain biomarkers in acute heart failure.
There are certain issues that should be addressed before acceptance:
- It is very intriguing that LVEF was not a predictor of events and rehospitalisation. It goes against every study on advanced heart failure which showed that LVEF is one of the most important predictors of events. The same with pharmacological treatment. No pharmacological treatment reduced the outcomes, which goes against a wide range of previous studies in acute heart failure.
- What was the etiology of AHF? What was the INTERMACS class? What was the mechanical circulatory support usage?
- Please do not use informal English language (“Table 1 shows the results of the main biochemical blood parameters (renal function, iron profile, haemogram...) in our population.”)
- Please cite PMID: 39333652.
Comments on the Quality of English LanguageCortés et al submitted an interesting manuscript entitled “New biomarkers in the prognostic assessment of acute heart failure with reduced ejection fraction: beyond natriuretic peptides” in which the authors evaluated certain biomarkers in acute heart failure.
There are certain issues that should be addressed before acceptance:
- It is very intriguing that LVEF was not a predictor of events and rehospitalisation. It goes against every study on advanced heart failure which showed that LVEF is one of the most important predictors of events. The same with pharmacological treatment. No pharmacological treatment reduced the outcomes, which goes against a wide range of previous studies in acute heart failure.
- What was the etiology of AHF? What was the INTERMACS class? What was the mechanical circulatory support usage?
- Please do not use informal English language (“Table 1 shows the results of the main biochemical blood parameters (renal function, iron profile, haemogram...) in our population.”)
- Please cite PMID: 39333652.
Reviewer 2 Report
Comments and Suggestions for Authors
The work titled " New biomarkers in the prognostic assessment of acute heart failure with reduced ejection fraction: beyond natriuretic peptides" by Cortes et al, is nicely drafted and well-articulated. The work provides a detailed and systematic exploration of the prognostic role of biomarkers in heart failure (HF), particularly in patients with reduced ejection fraction (HFrEF).
I have only minor concerns: In abstract avoid the use of the abbreviations. Please expand terms like LVEF (HFrEF), For better and more readability bars in the figures can be colored.
The discussion explains understanding prognostic markers in HFrEF, but its impact is diluted by dense writing, redundancies, and lack of clear segmentation. Addressing these issues would significantly enhance readability and accessibility.
The authors try to explore the new biomarkers for acute heart failure. GDF-15 and sST2 showed the highest predictive power for all-cause mortality, superior to natriuretic peptides as they improved prognostic assessment. The interleukin-1 receptor-like 1 (IL1RL1) protein, commonly referred to as ST2 (growth stimulation expressed gene 2), has emerged as a promising novel biomarker for AHF, the present work reinforces these findings.
The tables are well drafted and explain all the factors in detail. The references are recent and cover the topic addressed. For the methodology section, a flow diagram of steps is recommended for better understanding of the procedure followed from recruitment of patients till the assessment. The authors should discuss the prognostic and therapeutic response roles of GDF-15 and sST2 in HF, and how do they improve clinical decision-making and patient outcomes.
Reviewer 3 Report
Comments and Suggestions for Authors
- Figure 1: "all-cause of death" in the title is called "all-cause mortality" in the description, which should be corrected and better phrased "comparison of baseline characteristics according to clinical endpoint during follow-up (all-cause death)"
- Figure 1: the histogram for medical treatments lacks a specific description, therefore the reader cannot comprehend the 2 groups "light grey" vs "dark grey". I think a proper description should be added.
a distinction between home-medications vs in-hospital treatments may clarify the medical management of patients.
- lines 81 - 83: since the main focus of the study is the description of biomarkers, I think that a complete description or classification of them is required, with detailed specification of the reason why they were chosen as variables of interest
- Table 1: no proper Title and description of the table and its context.
- in Table 1: I think the concentrations of natriuretic peptides deserve a second check
- Figure 2: some elements are not clearly presented:
. HF: how should be interpreted the "HF variable" in the univariate Cox regression, when the sample study is composed of HF pts ?
. for continuous variables: the "change per unit" should be presented in the table
- Figure 3: the results of the multivariate Cox regression are poorly presented (no information regarding missing values, no AIC or BIC index as estimates of predictive errors, no equation provided for the model, no internal validation of the model)
- since no information is provided regarding the "time-to-event", why the authors decided to use Cox-regression and not logistic-regression ?
if time-to-event analysis is considered the endpoint of the study, can the authors provide a Kaplan-Meier graphic to show the estimated probability of survival / time-to-event ?
- Table 2: given the similarity between Table 2 and Figure 1, the authors should provide a standard visualization / description of baseline characteristics
- Table 4: although the Title shows "all-cause death" as the outcome variable of the regression, the authors have conducted the analysis for "HF readmission"
Comments on the Quality of English Language
- some minor orthographic errors in English
- repeated use of "etc." and "...", which limits the quality of the text and lowers the interest of the reader
Overall, the quality of English does not limit the understanding of the message of the article, but the description of statistical analysis and the choice of the predictors of the chosen regression methods are poor and should be implemented
Round 2
Reviewer 1 Report
Comments and Suggestions for Authors
The authors successfully addressed my comments.
Author Response
We wholeheartedly thank the reviewer for their prior comments.
Reviewer 3 Report
Comments and Suggestions for Authors
in Figure 4 there is an error in reporting the BIC value for the 2nd pairs of predictor (1.58.3 should be 158.3)
Lines 240 - 242: I think these lines deserve a revision.
